# Immature Teratoma: Diagnosis and Management—A Review of the Literature

**DOI:** 10.3390/diagnostics13091516

**Published:** 2023-04-23

**Authors:** Liviu Moraru, Melinda-Ildiko Mitranovici, Diana Maria Chiorean, Marius Coroș, Raluca Moraru, Ioan Emilian Oală, Sabin Gligore Turdean

**Affiliations:** 1Department of Anatomy, “George Emil Palade” University of Medicine, Pharmacy, Sciences and Technology, 540142 Targu Mures, Romania; 2Department of Obstetrics and Gynecology, Emergency County Hospital Hunedoara, 331057 Hunedoara, Romania; 3Department of Pathology, County Clinical Hospital of Targu Mures, 540072 Targu Mures, Romania; 4Department of Surgery, “George Emil Palade” University of Medicine, Pharmacy, Sciences and Technology, 540142 Targu Mures, Romania; 5Faculty of Medicine, “George Emil Palade” University of Medicine, Pharmacy, Sciences and Technology, 540142 Targu Mures, Romania; 6Department of Pathology, “George Emil Palade” University of Medicine, Pharmacy, Sciences and Technology, 540142 Targu Mures, Romania

**Keywords:** immature teratoma, therapeutical management, fertility preservation, germ cells, targeted treatment

## Abstract

An immature teratoma is a germinal malignant tumor composed of three germ cell layers, occurring more frequently in young women. It is the second most frequent among the malignant germinal tumors after dysgerminoma, and it is the only neoplasm with germ cells that are histologically graded. Even if we do not have a consensus regarding its therapeutical management, it has a good prognosis, with an excellent overall survival rate and good fertility preservation. More studies are needed regarding the necessity of adjuvant chemotherapy in pediatric oncology, and because of chemotherapy’s long-term adverse effects, surveillance or a targeted treatment is preferred, but the main therapy is fertility-sparing surgery. Special attention should be given to the genetic mapping of the histological pieces for patient risk stratification due to its value in prognosis and future treatment.

## 1. Introduction

An immature teratoma is a germinal malignant tumor that is composed of three germ cell layers (the ectoderm, endoderm, and mesoderm), and it is histologically characterized by immature tissue, most frequently, neuroepithelial tissue [1]. The most frequently affected individuals are adolescents and young adults in their reproductive age. It is the second most frequent, after dysgerminoma, among malignant germinal tumors [2]. An immature teratoma is the only neoplasm with germ cells that are histologically graded depending on the immature neural elements, and this is a prognostic factor for overall survival [1]. This grading system is called the Norris grading system [3]. Immature teratomas represent <1% of ovarian cancers, occurring more frequently in young women [4]. Statistically, its incidence is 2.3 per 100,000 patients [5]. It represents 35.6% of all germ tumors in young women.

The word teratoma comes from the ancient Greek “teraton”, which means monster [6]. It is a relatively little-known tumor because it is rare, and a small series of published studies have been undertaken [7]. There is no consensus on staging, and an agreement was made to combine FIGO staging with the TNM and COG (Children’s Oncology Group) [7]. In addition, we have taken into account that teratomas can be with mix component, containing other tumors with germ cells, such as a yolk sac, a mature teratoma, and dysgerminoma, and these other tumors can change a patient’s prognosis and make comparative studies irrelevant. In addition, pediatric cases and those involving adults must be studied separately as there are irreconcilable differences regarding the staging system and therapeutic management approaches, according to the MaGIC (Malignant Germ Cell Tumor International Collaborative). Pediatric cases have not received adjuvant chemotherapy except in rare situations, as there is disagreement regarding its usefulness compared to its long-term adverse effects [7]. In the case of adults in stage I, grade 1, a surveillance policy is already imposed rather than adjuvant chemotherapy [7].

## 2. Materials and Methods

We used the search engine Google Chrome to search publications from the PubMed, MEDLINE database, entering the keywords “immature teratoma therapeutical management, fertility preservation, germ cells, targeted treatment”.

We also queried the reference lists of high-cited articles for the selection of studies and considered eligible, articles written in English and published between 2012 and 2023. The inclusion criteria for the chosen articles were based on (a) classification, (b) metastases and evolution, (c) relation to pregnancy, (d) therapeutic management, (e) prognosis, and (f) follow-up. Other studies, which were written in languages other than English, published before 2012, or containing other mixed elements (mixed germ cell element content in tumors), were excluded from our data analysis.

Literature search found 6734 potentially eligible articles. Duplicates were excluded after the initial search. The others were excluded after full-text screening based on our exclusion criteria mentioned above or poorly designed studies (a lack of description of correct management or a coherently described diagnosis or case reports that were not relevant to our review).

### 2.1. PubMed Search Strategy

#### 6734 Results

((“teratoma”[MeSH Terms] OR “teratoma”[All Fields] OR (“immature”[All Fields] AND “teratoma”[All Fields]) OR “immature teratoma”[All Fields]) AND (2012:2023[pdat])) AND ((“teratoma”[MeSH Terms] OR “teratoma”[All Fields] OR (“immature”[All Fields] AND “teratoma”[All Fields]) OR “immature teratoma”[All Fields]) AND (“classification”[MeSH Terms] OR “classification”[All Fields] OR “classifications”[All Fields] OR “classification”[MeSH Subheading] OR “classification s”[All Fields] OR “classificator”[All Fields] OR “classificators”[All Fields]) AND ((“teratoma”[MeSH Terms] OR “teratoma”[All Fields] OR (“immature”[All Fields] AND “teratoma”[All Fields]) OR “immature teratoma”[All Fields]) AND (“metastasation”[All Fields] OR “metastasic”[All Fields] OR “metastasing”[All Fields] OR “metastasise”[All Fields] OR “metastasised”[All Fields] OR “metastasises”[All Fields] OR “metastasising”[All Fields] OR “metastasization”[All Fields] OR “metastasizes”[All Fields] OR “metastasizing”[All Fields] OR “neoplasm metastasis”[MeSH Terms] OR (“neoplasm”[All Fields] AND “metastasis”[All Fields]) OR “neoplasm metastasis”[All Fields] OR “metastase”[All Fields] OR “metastases”[All Fields] OR “metastasize”[All Fields] OR “metastasized”[All Fields]) AND (“biological evolution”[MeSH Terms] OR (“biological”[All Fields] AND “evolution”[All Fields]) OR “biological evolution”[All Fields] OR “evolution”[All Fields] OR “evolution s”[All Fields] OR “evolutional”[All Fields] OR “evolutions”[All Fields] OR “evolutive”[All Fields] OR “evolutivity”[All Fields]) AND ((“teratoma”[MeSH Terms] OR “teratoma”[All Fields] OR (“immature”[All Fields] AND “teratoma”[All Fields]) OR “immature teratoma”[All Fields]) AND (“pregnancy”[MeSH Terms] OR “pregnancy”[All Fields] OR “pregnancies”[All Fields] OR “pregnancy s”[All Fields]) AND (“teratoma”[MeSH Terms] OR “teratoma”[All Fields] OR (“immature”[All Fields] AND “teratoma”[All Fields]) OR “immature teratoma”[All Fields]) AND (“therapeutical”[All Fields] OR “therapeutically”[All Fields] OR “therapeuticals”[All Fields] OR “therapeutics”[MeSH Terms] OR “therapeutics”[All Fields] OR “therapeutic”[All Fields]) AND (“manage”[All Fields] OR “managed”[All Fields] OR “management s”[All Fields] OR “managements”[All Fields] OR “manager”[All Fields] OR “manager s”[All Fields] OR “managers”[All Fields] OR “manages”[All Fields] OR “managing”[All Fields] OR “managment”[All Fields] OR “organization and administration”[MeSH Terms] OR (“organization”[All Fields] AND “administration”[All Fields]) OR “organization and administration”[All Fields] OR “management”[All Fields] OR “disease management”[MeSH Terms] OR (“disease”[All Fields] AND “management”[All Fields]) OR “disease management”[All Fields]) AND ((“teratoma”[MeSH Terms] OR “teratoma”[All Fields] OR (“immature”[All Fields] AND “teratoma”[All Fields]) OR “immature teratoma”[All Fields]) AND (“prognosis”[MeSH Terms] OR “prognosis”[All Fields] OR “prognoses”[All Fields]) AND “follow-up”[All Fields]).

## 3. Epidemiology

Immature teratomas represent 10% of GCTs and 42% of malignant GCTs [8], although, as is shown in the introduction, according to other statistical data, they represent 35.6% of germinal tumors in young people [5]. Likewise, regarding its incidence, there are discrepancies of 2.3 per 100,000 patients [5] or 0.05 per 100,000 in the population [8]. These discrepancies are due to its rarity, which is such that a reliable statistic could not be attained. The affected age was less than 30 years old [9]. According to MOGCTs, mortality is higher in the African American population than in Caucasians. In addition, those with low incomes are more often diagnosed in the advanced stages of the disease [9].

## 4. Pathogenesis

The etiopathogenesis of these tumors is not clear. There are several hypotheses, such as their development from trapped oocytes following defective folliculogenesis [10]. They can also develop from pluripotent stem cells or residual fetal cells. They undergo a process of metaplasia under the influence of inflammatory factors that act as triggers [11]. However, these are all merely assumptions [11]. Their molecular and genetic etiopathogenesis is not known and remains unstudied [10]. The origin of immature teratomas has not yet been investigated. Mature teratomas are parthenogenetic tumors containing only maternal genomes. Mature and immature teratomas showed similar methylation levels, but there are some aberrant methylation levels in immature teratomas that differ from that in mature teratomas. DNA methylation is a mechanism of genome imprinting, and this situation suggests different pathogenic mechanisms. Such lesions correspond to mixed germ cell tumors [12], and immature teratomas are a subtype of malignant germ cell tumors of the ovary [13]. Diverse errors are the key to the formation of immature teratomas, and epigenetic differences are responses key to the variation in differentiation patterns in teratomas and for the transformation from benign to malignant tumors [13].

Multiregional exome sequencing of ovarian immature teratomas reveals a paucity of somatic mutations and extensive allelic imbalances, and a somatic mutation evaluation has been performed by The Cancer Genome Atlas Research Network. The existence of an extensive loss of heterozygosity and the absence of known oncogenic variants has been demonstrated. Heskett et al. in their study used genome-level sequencing to explain the pathogenic mechanism of immature teratomas for the first time and highlighted that meiotic nondisjunction events producing the 2Nnear-diploid genome and allelic imbalance are responsible for the arise of immature teratomas [13].

Homozygosity is well-known in mature teratomas, but the zygosity in immature teratomas is not well studied [14]. While a mature teratoma develops from the oocyte after the completion of meiosis I, with the failure of meiosis II, containing homozygous chromosomes, an immature teratoma has a different pathogenic pathway [14]. However, in immature teratomas, compared with mixed germ cell tumors, fewer genetic alterations were found.

The difference in the rate of homozygosity between pure immature teratomas and mixed germ cell tumors suggests different pathogenic pathways and biological behaviors. Pure immature teratomas harbor fewer detectable genetic alterations and lack the somatic abnormalities seen in mixed germ cell tumors. Further research is needed to provide additional insights into the pathogenesis of immature teratomas [14].

## 5. Methods of Diagnosis

As for the methods of diagnosing these ovarian tumors, according to data from the literature, several elements can help, such as biomarkers, clinical manifestations of the disease, or imaging data, but a definitive diagnosis is histological, with surgical staging.

(A)Signs and symptoms

The symptoms are not specific. Clinically, a pelvic tumor mass with pain is more frequently observed. Symptoms can become acute in the case of complications and include rupture, torsion, hemorrhage, and superinfection with peritonitis [6,8,15,16,17]. However, the specificity of these manifestations is low, and they also accompany other ovarian tumors [16,17];

(B) Biomarkers

Both in children and adults, modified markers appear, and here, we discuss CA-125, AFP, and beta-HCG, though not to the same extent as those in other germ cell tumors. Alpha-fetoprotein levels are elevated (AFP) but not above 1000 ng/mL [1,8]. However, these markers can also have normal values [15]. In children, the use of biomarkers has been abandoned, considering that values above 100 ng/mL are correlated with the presence of yolk sac elements [5]. This fact emphasizes the need for surgical intervention to make a correct diagnosis;

(C) Imaging

Ultrasonography is usually used, CT scans, and MRIs are used in cases where the immature teratoma imaging result is less specific, but these methods are useful in making a therapeutic decision. A mixed appearance, both solid and cystic, with calcifications may be presented [8,18]. More recently, 18-FDG PET has been used, which, compared to a CT scan, is superior for visualizing lymph nodes, and it is used for locating samples (the locations of areas to be biopsied). This method helps to choose the correct therapeutic strategy and is better for staging, monitoring, and detecting recurrences or metastases, but it is expensive. It is preferable where CT and MRI have failed [4,18];

(D) Anatomy and Histology

The ovaries are paired organs, representing the female gonads, located in the peritoneal cavity on the side wall in a depression called the ovarian fossa on one side and the uterus on the other. The peritoneum on the surface is missing, being replaced by a single-layered germinal epithelium under which a tunic is located (i.e., the albuginea). This tunic comes into contact with the cortical, a glandular portion that comes into contact with the medullary, which is constituted by loose connective tissue. The medulla is crisscrossed by blood vessels, lymphatics, and nerve fibers. In the cortex, the ovarian follicles develop in various developmental stages, and they are included in the fibrous conjunctival stroma. Macroscopically, it appears as a soft yellow tumor [18];

(E) Histopathology

Histopathologically, immature teratomas form as islands of poorly differentiated (primitive), immature, and blast-like cells. In most immature teratomas, the neuroectodermal elements are immature and dominant, and these are the easiest elements to recognize and quantify. Immature neuroectodermal tissues include islets and nests of neuroblasts, hypercellular and mitotically active immature glia, and primitive, melanic pigmented retinal tissue. Occasionally, there may be associated benign vascular proliferations associated with neural elements, it can be confused with a vascular tumor [19,20,21].

Macroscopically, an immature teratoma is predominantly or entirely solid and, typically, unilateral, and its dimensions are variable, with a typical diameter of approximately 15 cm, which is nearly double the average diameter of a mature teratoma (Figure 1).

On a section, the solid areas appear as white, copper, gray, or brown, and the consistency of the tissue is variable (soft or firm) where cartilage or bone is present. The contralateral ovary to the one with an immature teratoma can present a dermoid cyst in 7.1% of cases [22].

Microscopically, immature mesodermal tissue is hypercellular, consisting of spindle-shaped, small cells with hyperchromatic nuclei (Figure 2), which are mitotically active, and they may contain the foci of immature cartilage tissue, immature adipose tissue, osteoids, or even rhabdomyoblasts. Endodermal tissues are usually less represented and include primitive glands lined by columnar cells with vacuoles, resulting in an enteroblastic appearance. Immature renal (metanephrogenic) tissue is another rare type of immature tissue found in immature teratomas. The differential diagnosis of immature teratomas is made with mature teratomas, mixed germinal tumors, and mixed mesodermal tumors. Differentiating between a mature and an immature teratoma can be difficult, as a teratoma that has only a small amount of immature tissue is difficult to frame. It is recommended that these tumors be referred to as mature teratomas with microscopic foci of immature tissue [19,22,23,24].

In some immature teratomas, the microscopic foci of yolk sac tumors are observed. As long as these rare foci are less than 3 mm in diameter, they do not appear to have a negative impact on prognosis, and they do not justify the diagnosis of a mixed germ cell tumor. Extraovarian tumor implants located in the omentum, peritoneum, and lymph nodes are more frequently detected in patients with immature teratomas and only occasionally in those with mature teratomas. In the literature, approximately 100 such cases were detected. Immature tissue may be present in an implant, being mainly made up of mature neural tissue (Figure 2), which causes the implant to be designated as a gliomatosis. Although astroglia is the main component, the deposits are made up of glial cells and, invariably, neurons, neurofilaments, and other benign mesenchymal and epithelial elements are also present [23,24,25].

As an example, we present some macroscopic and microscopic images of a new immature teratoma case (Figure 1 and Figure 2). It is a 23-year-old nulliparous patient presenting with an abdominal tumor accompanied by gastrointestinal disorders. A CT scan confirmed the tumor, and it was followed by surgical intervention in order to remove the adherent tumor of the right ovary to the uterus and bladder on 24 December 2021. Adjuvant chemotherapy was indicated in this situation.

## 6. Classification

### 6.1. Types of Teratoma

Ovarian teratomas are of three types: mature, immature, and monodermal. A mature teratoma, also called a dermoid cyst, represents 95% of all teratomas and is benign. It has dimensions of 7–15 cm, with well-differentiated tissue displaying elements from all three layers (ectoderm, endoderm, and mesoderm). It is unilateral, cystic, rarely septate, and sometimes displays a protuberance called the Rokitansky nodule. It contains bone tissue, hair, teeth, sebum, fatty tissue, and keratin. An immature teratoma, which is the malignant form of teratomas, is the second most common type of teratoma. Its incidence rate is 0.05/100,000 of the population. In 80% of cases, it is unilateral, affecting mainly children and women of childbearing age. It has larger dimensions than a mature teratoma, and it can be solid or mixed with cystic elements. It contains sebaceous, mucinous, and serous fluids. Microscopically, the three immature layers with embryonic elements will often contain neuroepithelium, which gives it its degree of malignancy and represents an unfavorable prognostic factor [8].

The third type is a monodermal teratoma with a single constitutive histological element in the form of struma ovarii, a neuroectoderm, and a carcinoid. Ovarian strumae appear more frequently over the age of 40 and are more frequently benign, showing columnar epithelial cells and follicles. A neuroectoderm is more often primitive and malignant, affecting women aged 10–30 years, and it is formed by primitive brain tumors. Differentiated neuroectodermal tumors are less common. Carcinoid tumors occur more frequently during menopause, and they have four subtypes: insular, trabecular, mucinous, and stromal. Clinically, all are manifested by a pelvic–abdominal tumor mass with pain that can become acute in cases with complications such as rupture, torsion, hemorrhage, and superinfection with peritonitis. Imaging by ultrasonography, CT scans, and MRIs are standard, though, in the case of an immature teratoma, they are less specific, but they contribute, together with the clinical manifestations, to decision-making about therapeutic management. Than we confirm diagnosis histologically [2,8].

Teratoma-related tumor biomarkers are not as modified as those of other germinal tumors; instead, anti-N-methyl-D-aspartate (NMDA) can appear, which is a marker of immunological encephalitis [26] and is clinically associated with the presence of neurological effects, including seizures and comas. In the case of struma ovarii, hyperthyroidism appears, and the carcinoid releases serotonin with flashing, hypertension, and bronchospasm.

### 6.2. Extragonadal Teratoma

Extragonadal teratomas are rare, appearing on the midline and affecting the endometrium, cervix, and vagina. They can be associated with gliomatosis of the peritoneum, and after chemotherapy, they can develop growing teratoma syndrome (GTS) [11,18]. Such teratomas usually develop in the gonads, but they can also appear extragonadally on the germ cell migration route in 1–2% of primitive germinal tumors, developing from residual fetal cells or implantations at their level [11]. Thus, four hypotheses regarding their origin are taken into account: displaced germinal cells, derivation from pluripotent stem cells, the metaplastic process, and residual fetal tissue, as inflammation is present in their pathogenesis [11].

### 6.3. Growing Teratoma Syndrome

This is a syndrome that appears after chemotherapy for germ cell tumors without elements of malignancy. Herein, reference [13] included 26 IT patients with 69 months of follow-up, where 15 underwent chemotherapy and 6 developed GTS. The studied patients were between 17 and 38 years of age, with an average age of 24 years. Those with GTS required additional surgery, and no malignant elements were found in the histopathology. The areas of localization were the urinary bladder, peritoneum, contralateral ovary, abdominal wall, mediastinum, Douglas pouch, diaphragm, liver, omentum, and sigmoid [11,27,28]. In another study, out of 175 IT cases, 35 developed GTS. The interval from initial diagnosis to GTS was 18.5 months (6–78) [28]. In some cases, four reinterventions were necessary. Its occurrence was not related to the patient’s age, tumor grading, mature IT elements, or residual tumors after the first resection, according to this study. This fact was contradicted by a study of 35 cases of GTS [28] in which the main factor was residual disease after the first surgical intervention and gliomatosis of the peritoneum. It was first described by DiSaia, who reported GTS in three women after adjuvant chemotherapy for malignant IT [27]. The pathogenesis of GTS is controversial as chemotherapy is supposed to induce cell differentiation into mature tissue. According to another theory, chemotherapy selectively eliminates malignant tumor cells, which is why it is called therapeutic conversion. Tumor markers normalize in cases of GTS [27,28].

The evolution of GTS is unpredictable. The recommended therapy is surgical, with the need for effective debulking and, preferably, fertility-sparing surgery [27,28]. A multidisciplinary team is preferred as it is a pathology with a risk of hemorrhage [28]. It has the potential to transform into sarcomatous, adenocarcinoma, or neuroectodermal or carcinoid tumors, even after a long time since its appearance. If it is unresectable in terms of location, alpha-interferon therapy can be tried, but additional studies are necessary [27,28].

### 6.4. Gliomatosa Peritoneii

Peritoneal gliomatosis is defined by the presence of mature glial tumor tissue in the peritoneum. Its exact etiopathogenesis is not known, but it can be assumed that it is secondary to IT rupture. However, it was found that glial tissue is more reactive and genetically different from a primitive ovarian tumor, which raises questions [7].

## 7. Metastases and Unpredictable Evolutions

The most common form of spread is the hematogenous route, especially in the liver, retroperitoneum, omentum, mediastinum, and brain [15]. Metastasis in the small intestine is more unspecific, with biomarkers having normal values (CA-125, beta-HCG, and AFP), which is often associated with GTS. This syndrome also occurs due to the fact that reconversion chemotherapy is used, which can suppress the immature elements of metastases, developing only mature ones that are later removed surgically [15]. A teratoma can be complicated by anti-NMDPP encephalitis (anti-N-methyl-D-aspartate receptor encephalitis), which is a rare paraneoplastic syndrome [26]. It requires surgical treatment of the teratoma and immunotherapy for the encephalitis with iv monoclonal antibodies, such as rituximab, iv immunoglobulins, steroids, and plasmapheresis. A diagnosis of encephalitis is made by determining the anti-NMDAP antibodies in blood. This association is sporadic [26].

## 8. Immature Ovarian Teratoma and Pregnancy

It is interesting to observe this association and the outcomes for the mother and fetus. This is also viewed through the lens of oncofertility, which has been emphasized lately, as this is a malignant tumor that occurs at a young age. An association with pregnancy is found in 0.07% of cases [16]. In the literature, a study was presented of 24 patients with different stages and degrees of the disease in which maternal age, gestational age, stage, grade, clinical, imaging, treatment, and mother/fetus prognosis were followed. For the management of the disease, a multidisciplinary approach is mandatory. The golden standard is unilateral oophorectomy in order to be able to maintain the pregnancy. A second look is indicated during the cesarean section in situations where the correct staging could not be performed during the first intervention. In stage Ia, grade 1, chemotherapy is not indicated, and in stage I, grades 2 and 3, it is controversial. Staging after oophorectomy, peritoneal washing, and careful inspection of the abdominal cavity is preferred. If chemotherapy is necessary, stages II and III are expected in the second trimester of pregnancy, when the risk of malformation is reduced, and platinum-based chemotherapy is preferred [16]. Ventriculomegaly, transient neutropenia, and hearing loss were found in the newborns in these situations, and additional data are needed [16]. The risk of transplacental metastasis is also unknown.

## 9. Therapeutic Management, Prognosis, and Follow-Up

There are discrepancies between the proposed therapies, and this fact is due to the rarity of such a tumor and its characteristics. There are no randomized trials with clear data in the literature [7]. Urgent cooperation between pediatric and adult oncology is necessary to establish a generally accepted staging and a reliable treatment strategy.

Up to the present, adjuvant chemotherapy has been associated with surgery in adults (except for standard IA), but it is not used to treat children [7]. The chemotherapy used with the best results was platinum-based chemotherapy (BEP (bleomycin–etoposide–cisplatin)). The number of chemotherapy courses is not yet well defined, but three courses are preferred. There are relapses, which may take the form of immature IT teratomas, other germinal tumors, mature teratomas, glioblastomas of the peritoneum, or GTS (growing teratoma syndrome). The treatment of a relapse is surgical, with additional histopathological examinations performed [7,29].

The most important prognostic factor is tumor grading, which is completed using the immature neuroepithelial component. Generally, the tumor stage, incomplete resection, and the presence of the mixed elements of a yolk sac are used as prognostic elements and overall survival indicators. Gliomatosis of the peritoneum may be associated relatively frequently, and it also has a good prognosis, unlike all the others [7].

Chemoresistivity is another little-discussed phenomenon. The exact reason for its occurrence is not known [5,7].

GTS, mentioned earlier and detailed in another subsection, appears to be the result of adjuvant chemotherapy and contains elements of mature teratomas [7].

Another fact that is not established is the standard duration of follow-up. It is known that in the case of dysgerminoma, this is every 3 months for the first 2 years, every 6 months for the next 3 years, and annually for the next 5 years, and then follow-up is extended to annual re-evaluation for the next 10 years. This is not established in the case of IT. We only know that the re-evaluation includes clinical examination, imaging, and tumor markers, with the latter having minimal relevance as it may be normal even in the case of relapse [29]. Regarding the approach to pathology in pediatrics [5], the grade, stage according to FIGO and the COG, age, symptoms, and AFP are important as prognostic factors, but as biomarkers, they were not included in the table as they may be normal. Their values were also excluded when >100 ng/mL, and these values can increase because of the presence of yolk sac tumors in the IT component. Minimally aggressive surgical intervention is preferred as a therapy. Chemotherapy is not performed even for grade 3 tumors because it is considered that it does not prevent relapse, and the adverse effects are unacceptably high. The preferred method is fertility-sparing surgery in pediatric IT, with complete resection to avoid tumor relapse. However, experience in such cases is limited [5]. 

According to data from the literature, it has been found that fertility-sparing surgery is the optimal therapy for adults as well. It is interesting to follow the fertility outcome [30]. In a study that included 46 patients, it was concluded that the bad prognostic factor in the advanced stages was due to residual disease [30]. Not everyone agrees with this conservative approach, which is why all adults with IT std I grade 1 received adjuvant chemotherapy after surgical intervention [21]. A comparison was made that included seven adults and two pediatric trials in a study that followed the literature extracted from the Malignant Germ Cell International Collaborative database. There was a lack of consensus regarding staging and therapeutic management. Another extremely important finding was that the 5-year disease-free survival and overall survival in children was better even though they did not undergo adjuvant chemotherapy, which was related by the authors more to the stage of the tumor and the grading at the time of diagnosis. Other smaller studies had the same result. However, these reviews declared certain limitations, including different staging systems, missing data, and a lack of explicit documents regarding responses to chemotherapy [21]. However, this extensive review drew attention to the problem of consensus regarding the management and follow-up of patients, both adults and pediatrics, and the need to establish a unitary therapy that brings maximum benefits to patients, both in terms of survival and preserving future fertility.

## 10. Discussion

An immature teratoma is a malignant ovarian tumor that begins from the three germinal layers [1]. It is a rare tumor that affects the female population during reproductive ages (or even earlier) [2], and it is seldom studied precisely for this reason. There are differences in the approaches of international bodies in terms of diagnosis, staging, and therapeutic management (the MaGIC, FIGO, the TNM, and the COG), and all of these organizations are attempting to reach a consensus [7,21]. Regarding diagnosis, there are no biomarkers or clear clinical or imaging elements that differentiate it from other ovarian tumors, with histopathology being the only definitive method for diagnosis [1,4,5,8,15,16,17,18].

A common conclusion of the studies and reviews found in the literature (Table 1) was that the main therapy is fertility-sparing surgery, which maintains both menstruation and reproductive function, though there is no consensus regarding combination treatment with chemotherapy [21]. The most effective chemotherapy was platinum-based chemotherapy [21]. Despite the involvement of various international organizations concerning staging, a clear consensus has not yet been reached using a common FIGO and COG design [7,21]. This, too, has limitations due to the possibility of the presence of a mixture of immature elements, more often from the yolk sac, which can also change the prognosis [5].

Another fact on which there is a consensus is that the most important prognostic factor is the tumor grade [7,21]. We note that a teratoma is the only malignant tumor with histologically graded germ cells [1,3].

There is no consensus regarding the follow-up period and the modality of a coherent follow-up [17,29], nor is there a coherent approach regarding relapse. What is now known is that chemotherapy has not proven its effectiveness in these cases, and thus, surgical intervention with tumor resection is likely the optimal approach [7]. Residual disease after intervention surgery appears to be the cause of GTS (growing teratoma syndrome), which is also resistant to chemotherapy and radiotherapy [9].

A lymphadenectomy or lymph node biopsy and omentectomy, which are part of the comprehensive surgical staging of malignant germ cell tumors in adult patients according to the National Comprehensive Cancer Network (NCCN), may not be recommended and performed due to a good prognosis as they are considered unjustified aggressive treatments that could potentially affect a patient’s quality of life [9].

The correct approach in the case of resistance to chemotherapy is not known. Various targeted treatments that have been proposed are still being studied. The chromosomal mapping of the histological piece could be useful both in the evaluation of other family members regarding the possibility of the appearance of the pathology and in terms of finding an alternative targeted treatment [1,7,10,31].

Consensus regarding the management of these tumors is needed for a multidisciplinary approach in a specialized clinic [16].

Even though this subject has been studied in the specialized literature, there are limitations in these studies. They were retrospective, there are few cases of these tumors, there exists a misdiagnosis risk, and there was no consistency in what was followed, which data were collected, and the accuracy of the results.
diagnostics-13-01516-t001_Table 1Table 1Comparison between studies: management and therapy approaches.**Article****No. of Patients****Median Age and Range****Grade****Stage****Management****Follow-Up****DFS****OS****Fertility**[1] **Alwazzan**27 patients>18 y.o.(18–42 y.o.)Grd. 1: 9 patients (33%)Grd. 2: 3 patients(11%)Grd. 3: 15 patients(56%)FIGO▪ I: 82%▪ II: 11%▪ III: 7%Surgery: 100%Radical hysterectomy with BSO: 11%Enucleation: 14%USO: 85%78%60 monthsOne relapse (grade 3) that had complete clinical response after chemotherapy
12 preg.10live births2 misc.[24] **Wang**46 patients22 y.o.(18–40 y.o.)▪ Grd. 1: 11 patients (23.91%)▪ Grd. 2 and 3: 35 patients (76.08%)▪ II: 16 patients (34.78%)▪ III: 30 patients (65.21%)Surgery: 100%Fertility-sparing: 38Radical: 8ChemotherapyAll: 100%74.2 months5 y.: 69.1%5 y.: 89.9%5 successfulpregnancies[25] **Pashankar**98 pediatric patientsPediatric patients12 y.o.(6–17 y.o.)▪ Grd. 1: 31%▪ Grd. 2: 20%▪ Grd. 3: 39%▪ Unknown: 10%COG▪ I: 60%▪ II: 12%▪ III: 28%▪ IV: 0%100%8 patients(8%)5 y.5 y.: 91%5 y.: 99%Unknown81 adult patientsAdults22 y.o.(18–42 y.o.)▪ Grd. 1: 9%▪ Grd 2: 33%▪ Grd 3: 56%▪ Unknown: 2%FIGO▪ I: 53%▪ II: 6%▪ III: 33%▪ IV: 7%100%100%5 y.87%93%Unknown[32] **Terenzani**69 patients7 months(0–12 months)Grd. 1: 25 patients(36.23%)Grd. 2: 25 patients(36.23%)Grd. 3: 11 patients (15.9%)Unknown: 8 patients(11.6%)Not given(extragonadal sites—sacrococcyx: 55%)All: 100% enucleationChemotherapy4 cases(5.79%)36 monthsAlways: 9 (13%)local in all casesChemotherapy-associated in 1 case; with surgery, in another case with radiotherapySurgery: 3 (4.34%) glioblastoma peritonei100%Spared in all cases[13] **Salmaz Hasdner**IT and pregnancy-24 patients27 y.o.(22–32 y.o.)▪ Grd. 1: 4 patients(16%)▪ Grd. 2: 5 patients(24%)▪ Grd. 3: 8 patients(32%)▪ Unknown: 7 patients(28%)▪ I: 11 patients(47.8%)▪ II: none▪ III: 4 patients(17.3%)▪ IV: none▪ Unknown: 8 patients(34.8%)Surgery: 100%C-section: 52.6%Teratoma rupture: 17.4%Miscarriage: 10.5%6 months

83.4%(4 patients rapidly died)19 newbornsMiscarriage: 10.5% 1 newborn with intracranial IT,1 with hypospadias (prematurity complication)3 subsequent preg.[9] **Jorge**1045 patients>18 y.o.(18–42 y.o.)▪ Grd. 1: 22.1%▪ Grd. 2: 23.8%▪ Grd. 3: 37%▪ Unknown: 17%FIGO▪ I: 69.5%▪ II: 6.5%▪ III: 13.7%▪ IV: 2.4%Surgical: 100%USO: 52.5%BSO: 9.3%Omentectomy: 27.4%Debulking: 8.8%Unknown: 4.4%ChemotherapyChemotherapy: 56.8%None: 38.8%Unknown: 4.4%5 y.
Std I: 98.3%Std II: 93.2%Std III: 83.7%Std IV: 72%Unknown[5] **Shinkai**7 patients7–13y.o.9 y.o.Grd. 1: 1 patient (14.2%)Grd. 2: 3 patients (42.9%)Grd. 3: 3 patients (42.9%)▪ I A (COG): 100%Surgery: all (100%)Nucleation: 2USO: 5ChemotherapyNone7 y.100%100%
**FIGO**—The International Federation of Gynecology and Obstetrics; **COG**—Children’s Oncology Group; **IT**— immature teratoma; **y.o.**—years old; **y.**—years; **preg.—**pregnancies; **USO**—unilateral salpingo-oophorectomy; **BSO**—bilateral salpingo-oophorectomy; **DFS**—disease-free survival; **OS**—overall survival.

## 11. The Future in Teratomas

The genetics of these tumors are being studied, and they show a common cellular origin from germ cells in various stages of development but with different pathogenic pathways and different behaviors [31,33]. Its importance is also therapeutic because it has been observed that resistance to chemotherapy has a genetic component [10,33]. Epigenetic differences are also important; this may be the only known human tumor type where epigenetic dysregulation is responsible for the transformation from a benign to a malignant tumor [13]. In addition, multiple genetic alterations lead to poor prognosis and risk of recurrence [14], hence, the need for genetic mapping of the histological pieces to stratify the patient’s risk.

The optimal management for relapse needs to be determined, as does the question of whether chemotherapy has a place here. [7]. However, the most important element is the initiation of studies regarding targeted therapy [7]. Although it has been established that fertility-sparing therapy is optimal, in less fortunate cases egg donation has been successful. Contamination with malignant cells is, for now, theoretical. It is a solution for patients who would later require the removal of both ovaries. In addition, for such patients, gonadotropin therapy is controversial (a gonadotropin-releasing hormone agonist) as it can lead to premature ovarian failure in young patients, especially in the case of gonadotoxic chemotherapy. Another solution is the preservation of the ovarian tissue, but this is an invasive method, and in most countries, it is still an experimental procedure [1]. These are continuing concerns in current oncofertility.

## 12. Conclusions

An immature teratoma is a rare germinal malignant tumor, and it is the only such tumor for which we have tumor grading among germ cell tumors and for which, although we do not have consensus regarding the management of the pathology, there is a good therapeutic outcome, along with good survival rates and a high degree of preservation of fertility. Additional studies are necessary to clarify the need for chemotherapy, especially in pediatric oncology. Surveillance is preferred, taking into account chemotherapy’s adverse effects. Also possible alternative targeted treatments represents the future. These efforts are necessary due to the age at which these malignant tumors appear and the potential to cure the disease. Special attention should be given to the genetic mapping of the histological pieces for patient risk stratification due to its value in prognosis and future treatment.

## Figures and Tables

**Figure 1 diagnostics-13-01516-f001:**
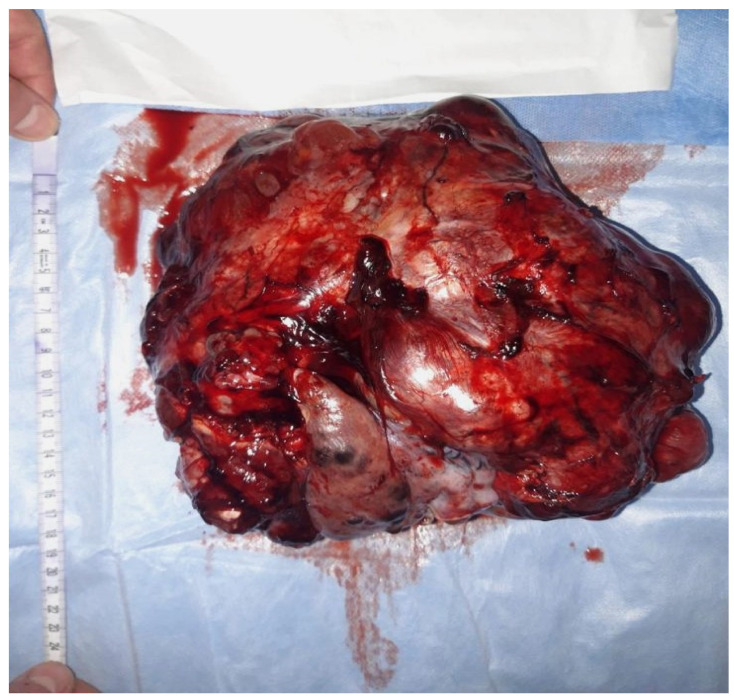
Macroscopic aspects of a surgical specimen from a 23-year-old nulliparous patient presenting with an abdominal tumor accompanied by gastrointestinal disorders. A surgical intervention was performed in order to remove the adherent tumor of the right ovary to the uterus and bladder on 24 December 2021.

**Figure 2 diagnostics-13-01516-f002:**
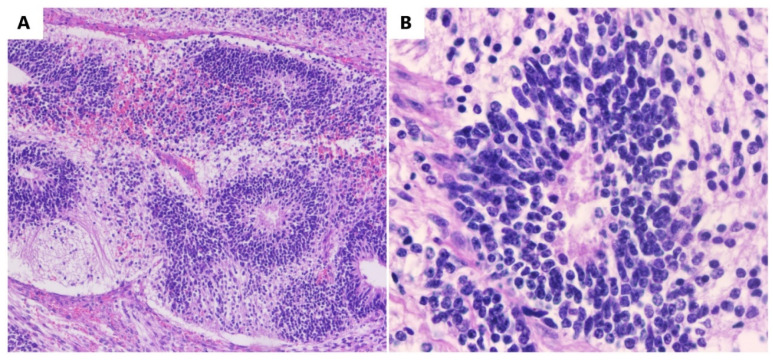
Immature teratomas. (**A**) Immature neuroectodermal tissue with rosettes (HE, ob. 10×). (**B**) Detail of the neuroectodermal tissue (HE, ob. 40×).

## Data Availability

Not applicable.

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
