# Peer review of "Immature Teratoma: Diagnosis and Management—A Review of the Literature"

_diagnostics, 2023, doi:10.3390/diagnostics13091516_

Round 1

Reviewer 1 Report

In this review article, the authors present an overview of the diagnosis and management of immature teratoma.

This paper could be a valuable contribution since there doesn’t appear to be recent review articles on this topic.  Nevertheless, the present manuscript is poorly organized, and the review of the literature is incomplete.

The structure of the manuscript should be reorganized as follows:  1. Introduction, 2. Materials and methods, 3. Epidemiology, 4. Pathogenesis, 5. Methods of diagnosis, 6. Classification, 7. Metastases and unpredictable evolutions, 8. Immature ovarian teratoma and pregnancy, 9. Therapeutic management, prognosis and follow-up, 10. Discussion, 11. The future in teratomas, 12. Conclusion.

Line 57-61: describe the literature search strategy in more detail.  How many articles were retrieved using PubMed?  How many were retained? How many were excluded (not pertinent articles, poorly designed studies. duplicate, etc.)?

Line 71-77: the review of the pathogenesis of immature teratoma is very inadequate.  Expand the review of that literature and the discussion of this topic.  Include information about the genetic and molecular basis of immature teratoma, such as the articles cited below.

Line 82-94: change the order of the paragraphs: A. Signs and symptoms, B. Biomarkers.

Table 1: organize the cited studies in some logical order, such as 1. Pediatric immature teratoma, 2. Adult immature teratoma, 3. Pregnant adult immature teratoma.  Alternatively, present the studies in alphabetic order of the author’s name or in chronological order.  Change “Medium age” to “Median age and range”.  The number of patients don't add up: for Hasdner study, you mention 24 patients, but 4+6+8+7 = 25 patients.  

Line 456-457: why do you write “Written informed consent has been obtained from our patient to publish this paper”?  Typically, no informed consent is need for a retrospective review of the literature.  Are the pathology images (Fig. 1 and 2) from a new teratoma case? At the least, you need to present key information (age, pregnancy status, tumor location, etc.) about the patient.

References

References 18 and 25 are duplicate citations of the same paper.

Carefully review the literature to include pertinent articles, such as

Pashankar F, et al. Addressing the diagnostic and therapeutic dilemmas of ovarian immature teratoma: Report from a clinicopathologic consensus conference. Eur J Cancer. 2022 Sep;173:59-70. doi: 10.1016/j.ejca.2022.06.006.

Terenziani M, et al. Mature and immature teratoma: A report from the second Italian pediatric study. Pediatr Blood Cancer. 2015 Jul;62(7):1202-8. doi: 10.1002/pbc.25423.

Heskett MB, et al. Multiregion exome sequencing of ovarian immature teratomas reveals 2N near-diploid genomes, paucity of somatic mutations, and extensive allelic imbalances shared across mature, immature, and disseminated components. Mod Pathol. 2020 Jun;33(6):1193-1206. doi: 10.1038/s41379-019-0446-y.

Kato N, et al. Mature and immature ovarian teratomas share methylation profiles of imprinted genes: a MS-MLPA analysis. Virchows Arch. 2023 Jan 13. doi: 10.1007/s00428-023-03491-z.

Snir OL, et al. Frequent homozygosity in both mature and immature ovarian teratomas: a shared genetic basis of tumorigenesis. Mod Pathol. 2017 Oct;30(10):1467-1475. doi: 10.1038/modpathol.2017.66

Author Response

Cover letter:

Dear reviewer,I reorganized the structure of the manuscript as you sugested.

Line 57-61 became 66-118 where I detailed the literature search strategy.

Line 71-77 I add in the pathogenesis section from exemplified references very useful details related to genetics, for which I thank you, as we do not have genetician in our team. I changed the discussions and future management accordingly.[line 136-150,676-680]

Line 82-94 I changed the order: A .Signs and symptoms, B. Biomarkers.

Table 1. I organized the table in alphabetic order of the authors name. Here I kept in the same table the adult cases , pediatric and pregnant women associated with immature teratoma to have an overview of the differences in staging, possible management and evolution in these situations. I made all the changes you suggested. And you are right , in Hasdner study there are 24 patients, there was a mistake in the article that I found and corrected, grade 2 tumor found in only 5 patients instead of 6.

Line 456-457 the pathology images are from a new immature teratoma case. As an example , we present some macroscopic and microscopic images of a new immature teratoma case. It is a 23-year-old nulliparous patient presenting for an abdominal tumor accompanied by gastrointestinal disorders. CT scan confirms the tumor and it is followed  by a surgical intervention in order to remove the adherent tumor of the right ovary to the uterus and bladder on December 24,2021.Adjuvant chemotherapy was indicated in this situation.[line 283-289]

I have included the references in the manuscript and I removed the duplicate.

Thank you.

Reviewer 2 Report

A paper sent to me for evaluation analyzing the difficult problem of Immature Teratoma a germinal malignant tumor. I read the manuscript with great interest. In my opinion, the discussion section is edited too laconically. There is a lack of clearly formulated conclusions both, in the main text of the paper and in the accompanying abstract.  The manuscript touches on an interesting topic but the shortcomings of the discussion section and the lack of readable conclusions detract from the merit of the work. In my opinion, these editorial deficiencies are worth correcting. 

Author Response

Cover letter:

Dear reviewer ,I reorganized the structure of the manuscript , I hope now is more clear. I added in section The future in teratomas some ideas which bring improvements. I made additions to the abstract, pathogenesis, discussions and conclusions .Thank you for your suggestions!

Round 2

Reviewer 1 Report

Thank you for the improvements you made to your manuscript. 

Line 64-74: the keywords used are not just “immature teratoma” but all those listed at line 79-115.  Please put together.  Why did you limit your search to publications since 2012?  Please justify.  If possible, expand your search.  Also, explain what you mean by “mixed elements” and “poorly designed studies”.  What made a study well designed?

Line 127-147: please clarify the description of the genetic pathogenesis of immature teratomas.  Please seek assistance from knowledgeable specialists, as needed.

Line 353: remove “the guarantee of no recurrence”.

Line 638-641: the statement “this being probably the only human tumor type where epigenetic dysregulation is responsible for the transformation” should be rephrased to say that “this may be the only known human tumor type…”

Figure 1: add below the figure title a legend that summarizes the case illustrated.  Please expand on the information included in your response letter.

Table 1: for each article, indicate specific median age and range of ages of the patients.

Line 723-724: revise the informed consent statement as follows: “No informed consent was needed for the retrospective review of the literature.  For the case of immature teratoma presented in the figures, written informed consent was obtained from the patient.”

Author Response

Cover letter:

Dear reviewer

Line 63-64,71-72,75-77.The important articles published before 2012 are already cited in the review type manuscripts used by me in the article. By  „mixed elements „ I meant mixed germ cell elements content in tumors , I intended to includ only pure immature teratomas in my manuscript.I explain in the text by poorly designed study I mean those studies with a lack of description of a correct management or a coherently described diagnosis, or case reports that are not relevant to our review.

Pathogenesis:Line 144-151:Multiregion exome sequencing of ovarian immature teratoma reveals paucity of somatic mutations, extensive allelic imbalances, somatic mutation evaluation has been performed by The Cancer Genome Atlas Research Network.It has been demonstrated the existence of extensive loss of heterozygosity and the absence of known oncogenic variants.M.B.Heskett et al. in their study have used for the first time genome level sequencing to explain the pathogenic mechanism of immature teratoma and highlighted that meiotic nondisjunction events producing 2Nnear- diploid genome and allelic imbalance are responsible for the arise of immature teratoma.[13]

Pathogenesis:Line:158-162The difference in the rate of homozygosity between pure immature teratomas and mixed germ cell tumors suggests differetnt pathogenic pathways and different biological behaviors.Pure immature teratoma harbor fewer genetic alteration detectable and lacked somatic abnormalities seen in mixed germ cell tumors. Further study needed to provide additional insight into the pathogenesis of immature teratomas.

Line 374 :I removed „guarantee for no recurrence”

Line 657-659: I rephrased „this may be the only known human tumor type”

Figure 1: I added a legend based on the cover letter

Table 1. I added a mean age and range of ages to each article

Line 741-743 I revised the informed consent statement as follows:”No informed consent was needed for the retrospective review of the literature.For the case of immature teratoma presented in the figures , written informed consent was obtained from the patient.”

For these ,already existing track changes activated, I could not highlight the new changes.

And about the English translation, I used MDPI editing services.

Thank you.

Reviewer 2 Report

In my opinion, the paper in its current form deserves to be published in the Diagnostics journal.

Author Response

Thank you very much